# Neurological Manifestations of SARS-CoV2 Infection: A Narrative Review

**DOI:** 10.3390/brainsci12111531

**Published:** 2022-11-12

**Authors:** Bogdan Pavel, Ruxandra Moroti, Ana Spataru, Mihaela Roxana Popescu, Anca Maria Panaitescu, Ana-Maria Zagrean

**Affiliations:** 1Department of Functional Sciences, Carol Davila University of Medicine and Pharmacy, 050474 Bucharest, Romania; 2Clinical Emergency Hospital of Plastic, Reconstructive Surgery and Burns, 010713 Bucharest, Romania; 3Clinical Department 2, Carol Davila University of Medicine and Pharmacy, 050474 Bucharest, Romania; 4Matei Bals National Institute of Infectious Diseases, 021105 Bucharest, Romania; 5Department of Critical Care, King’s College Hospital, Denmark Hill, London SE5 9RS, UK; 6Cardiothoracic Medicine Department, University of Medicine and Pharmacy Carol Davila, 020021 Bucharest, Romania; 7Department of Cardiology, Elias Emergency University Hospital, 011461 Bucharest, Romania; 8Department of Obstetrics and Gynecology Filantropia Clinical Hospital Bucharest, 011171 Bucharest, Romania; 9Department of Obstetrics and Gynecology, Carol Davila University of Medicine and Pharmacy, 050474 Bucharest, Romania

**Keywords:** coronavirus, COVID-19, SARS-CoV-2, neurological impairment, viral encephalitis, coma

## Abstract

The COVID-19 virus frequently causes neurological complications. These have been described in various forms in adults and children. Headache, seizures, coma, and encephalitis are some of the manifestations of SARS-CoV-2-induced neurological impairment. Recent publications have revealed important aspects of viral pathophysiology and its involvement in nervous-system impairment in humans. We evaluated the latest literature describing the relationship between COVID-19 infection and the central nervous system. We searched three databases for observational and interventional studies in adults published between December 2019 and September 2022. We discussed in narrative form the neurological impairment associated with COVID-19, including clinical signs and symptoms, imaging abnormalities, and the pathophysiology of SARS-CoV2-induced neurological damage.

## 1. Introduction

COVID-19 infection can be associated with a considerable range of neurological disease, including fatigue, headache, encephalopathy, peripheral neuropathy, coma, neuropsychiatric symptoms, cerebro-vascular disease, and seizures. Numerous recent observational studies suggest that the new coronavirus is responsible for subjective neurological complaints and objective changes in the central and peripheral nervous system that result in distinct neurophysiological, imaging, and laboratory abnormalities. 

The frequency of new-onset neurological symptoms developing in patients with SARS-CoV-2 infection is reported variably in the literature. Espiritu et al. found a prevalence of 18.4% in a cohort of more than 10,000 patients in the Philippines [1]. In contrast, Garcia et al. found a much higher incidence (79.4%) of neurological manifestations in a retrospective cohort of Mexican patients [2]. It is likely that these differences are, at least partially, secondary to the subjective nature of some of the self-reported neurological symptoms. 

The mortality of patients with COVID-19 and neurological symptoms appears to be higher than the mortality of patients without neurological manifestation. In a recent systematic review, Mahdizare estimated a mortality of 29.1% in these patients [3]. Similar findings were published by the authors of the GCS-neuroCOVID Consortium and ENERGY Consortium, two large global consortia describing the neurological consequences of COVID-19 infection [4]. This study established that the presence of objective neurological signs was associated with 5.9 times higher odds of in-hospital mortality after adjustment for the hospital site, age, sex, race, and ethnicity [4].

The long-term outcomes after neuroCOVID are poorly explored. Long COVID, a syndrome characterized by a prolonged disease course, is frequently associated with neurological abnormalities. Persistent and long-lasting symptoms such as fatigue, headache, and attention disorders have been described as part of this entity [5]. An observational study showed that more than half of the patients who presented with neurological manifestations associated with SARS-CoV-2 infection had significant residual disability and impaired cognition [6]. In another study, 53% of the patients with neuroCOVID had a worse functional status at discharge [7]. Acute stroke, admission to the intensive care unit (ICU), and older age were predictors of a worse outcome [7]. The significant association between neurological manifestations and long-term outcomes underlies the importance of characterizing neurological impairment and recognizing its manifestations early in the disease course to inform therapeutic decisions in these patients. 

Although multiple observational studies have described the neurological manifestations of SARS-CoV-2 infection, systematic and unbiased data in this area are currently lacking. New initiatives have been created to maintain accurate records of neurological pathology post-COVID. The COVID-19 Neuro Databank-Biobank, for example, is a database currently gathering data on patients with neuroCOVID on a large scale [8]. The hope is that the availability of accurate data could enable practitioners and policymakers to make adequate decisions for the affected patients.

This review presents the newest developments in the area of SARS-CoV-2-induced neurological impairment, together with clinical and pathophysiological descriptors of the disease process. It is important for health professionals to have this information, given the high impact of this disease on the active workforce [9]. 

## 2. Materials and Methods

We searched MEDLINE, Cochrane Central Register of Controlled Trials (CENTRAL), and Embase between the 1^st^ of December 2019 and the 1^st^ of September 2022, without language restrictions. We included the following search terms: “coronavirus,” “COVID,” “COVID-19,” “SARS-CoV-2,” “corona,” “neuroCOVID,” “encephalopathy,” “neurological impairment,” “neurological deterioration,” “encephalitis,” “coma,” and “neurological symptoms.” We included both observational and interventional studies conducted in adults. We did not search grey literature (including commercial reports, white papers, and governmental publications).

We extracted from the included studies information regarding clinical manifestations, imaging changes, and disease pathophysiology. We referred to adult human subjects only for the description of clinical findings. Where appropriate, we also included studies of non-human subjects to illustrate pertinent pathophysiological findings. 

We reported the results in narrative form under three headings: clinical manifestations, neuroimaging, and pathophysiology. We also described patient-related clinical outcomes where available.

## 3. Neurological Impairment in COVID-19

### 3.1. Clinical Manifestations

#### 3.1.1. Nonspecific Neurological/Neuropsychiatric Symptoms 

Fatigue, headache, dizziness, depression, anxiety, and confusion are the most frequent non-specific symptoms in patients with COVID-19 during acute illness, but also prevail as long-term symptoms after disease resolution. Attention disorder, sleep abnormalities, mood disorders, memory loss, and post-traumatic stress disorder (PTSD) have been mostly described as long-term sequelae. 

Fatigue, although likely underdiagnosed due to its subjective nature, is a very common symptom during acute illness, its prevalence being estimated at 27% to 32% [10,11]. After COVID-19 resolution, fatigue is the most commonly reported symptom, appearing in up to 58% of patients in a systematic review that included 47,910 post-COVID-19 cases [5]. In a study in which patients were interviewed at a median of 10 weeks after COVID-19, 52% of the respondents reported persistent fatigue on a standardized questionnaire. This was not correlated with the severity of the viral illness or with the changes in biological markers during the acute phase. However, there was a correlation with female gender and preexisting depression/anxiety disorders [12].Headache was reported in 25% to 47% of cases during the acute COVID-19 phase [10,11,13]. After illness resolution, different reports suggested variable frequency of this symptom ranging from 10 to 44% [5,13].Dizziness or vertigo was reported in 18% to 26% of cases during acute illness, with a reduction to 3% after the viral infection resolved [5,10,11].Psychiatric disturbances were also described more frequently during the pandemic. In a retrospective cohort in the US, the incidence of psychiatric diagnoses in the first three months after COVID-19 infection was 5.8% in patients without any previous psychiatric history [14]. In this study, the most frequent disorders were anxiety, insomnia, and dementia [14].Depression and/or anxiety ranged from 10% to 38% during the acute infection and remained at 5%–13% after it resolved [5,10,11,14].Sleep abnormalities were inconstantly reported in the acute phase. A study using a simple yes or no questionnaire including 103 patients reported this symptom in 49% of the respondents [11]. In the long term, 11% of patients complained of persistent sleep disturbance after infection with SARS-CoV-2 [5].PTSD related to COVID-19 disease was reported in 1% of cases [5]. The common neurological features of long COVID-19, such as fatigue, brain fog, sleep disturbance, dizziness, anxiety, and depression, are also found in burnout and PTSD [15,16].Functional motor disorders (FMD) are a category of abnormalities characterized by impaired motor control and encompass an overlap of abnormal gait, myoclonus, jerking movements, tremor, or dystonia [17]. Frequently, fatigue and anxiety are also part of the disease spectrum [17]. Various authors reported an increase in the incidence of FMD during the COVID-19 pandemic, possibly in relation to a higher level of stress [18,19]. Hull et al. reported an increase of 60% in the number of patients diagnosed with FMD during the pandemic [18]. In a case-control study in Italy, about one third of patients had worsening of their motor symptoms and 18% had new-onset symptoms [20]. FMD has also been reported after COVID-19 vaccination, likely increasing hesitancy to vaccination in the general population [21].Psychological stress generated by the pandemic-related restrictions or alarming news in the media likely exacerbate pre-existing anxiety, panic, or depression disorders, thus potentially impacting the outcomes of numerous COVID-19 patients [22]. Our own observations are that most patients who manifest post-COVID neuropsychiatric disorders had pre-existing manifestations such as anxiety, depression, or sleep disturbance. These observations are in line with the findings of Hull et al. that reported a high prevalence of psychiatric disorders among patients with newly diagnosed FMD [18]. Interestingly, at the same time, a prior psychiatric diagnosis was an independent risk factor for developing COVID-19, as demonstrated by Taquet *et al.* in a large retrospective cohort [14].

#### 3.1.2. Central Neurological Disorders 

Encephalopathy was frequently reported in patients with COVID-19, with a prevalence of between 8% and 12% [2,3]. The term “encephalopathy” is non-specific and encompasses alteration of consciousness, confusion, delirium, agitation, or coma. A subset of patients that have a high incidence of encephalopathy associated with COVID-19 are critically ill adults. In those admitted to the ICU, agitation and delirium have an incidence of more than 60% [23]. An important and fatal manifestation of cerebral encephalopathy is acute cerebral edema. It has an estimated prevalence of 1% and was reported to cause tonsillar herniation and death in critically ill patients [3].Meningitis and encephalitis can present as neurological manifestations associated with COVID-19. A systematic review showed that the average time from the diagnosis of the viral infection to the onset of encephalitis was under 15 days [24]. Von Weyhern identified lymphocytic meningitis and encephalitis on autopsy in six patients that died of COVID-19 infection [25]. Half of these patients had normal neurology on admission to the ICU [25]. The exact pathophysiology of these disorders remains unclear, but a direct invasion of the neurological tissue, molecular mimicry, and systemic inflammation have been postulated as possible factors [24].New-onset seizures are also an important feature of neurological dysfunction post SARS-CoV-2 infection. A recent large observational study reported that acute seizure activity occurs in less than 5% in patients hospitalized with COVID-19 [10]. In this cohort, seizures frequently accompanied acute stroke or encephalitis. Data from the Registry of the Spanish Neurological Society suggest that most seizure episodes develop in the absence of pre-existent epilepsy [26]. Seizure activity in these patients is closely correlated with electroencephalographic abnormalities. Santos et al. reported electrographic seizures in 9.4% of patients admitted with acute SARS-CoV-2 infection, focal interictal epileptiform discharges in 18.8% of patients, and interictal continuum patterns in 25% of cases [27]. Clinical epileptic features include, in order of decreasing frequency, generalized tonic–clonic seizures, focal impaired-awareness seizures, status epilepticus, and secondary generalized seizures [26].COVID-19 infection was also associated with cerebro-vascular disease, including ischemic stroke, intracerebral hemorrhage, and intracerebral thrombosis. These complications have been described in old patients with multiple cerebro-vascular risk factors, as well as in young patients with no comorbidities. Mahdizare estimates a pooled prevalence of 2.9% of cerebral ischemia and 2.2% of cerebral thrombosis in adults with COVID-19 [3]. In a case series, more than two thirds of patients with stroke associated with SARS-CoV-2 had biochemical evidence of coagulopathy [28]. Cerebral thrombosis is a rare but severe manifestation on the neuroCOVID spectrum. The superior sagittal sinus is a frequent situs of thrombosis, accounting for two thirds of the events [29]. SARS-CoV-2 vaccination was also associated with arterial and venous thrombosis. Perry et al. found that patients with vaccine-induced thrombotic thrombocytopenia (VITT) had more intracranial thrombosis than those without VITT [30]. In this cohort, patients with VITT-associated cerebral thrombosis had a higher mortality or dependency status compared to the non-VITT control group (37% vs. 16%) [30].

#### 3.1.3. Peripheral Neurological Disorders

An increasing body of evidence suggests that COVID-19 can also affect the peripheral nervous system (PNS) in 1.3% to 9.5% of cases [31]. In a retrospective cohort including 1760 patients, 1.8% were diagnosed with PNS; in another large prospective cohort, 1.3% of patients developed PNS [32,33]. Cranial and peripheral neuropathies, Guillain-Barre syndrome (GBS), and myasthenia gravis have been described in these patients. 

GBS was also described to date in multiple patients infected with SARS-CoV-2. The majority presented with sensorimotor symptoms at an interval of 10 to 23 days after the diagnosis of the viral infection [31]. The incidence of GBS varied between 0.06% and 1% [10,31,32,33]. Overall, the clinical features and the clinical progression do not appear different from other post-infectious cases of GBS. Other less frequent PNS symptoms included variated mono- and polyneuropathies, and autonomic dysfunction [31,34].Myasthenia gravis was described rarely in patients with COVID-19. This association was mostly reported as case reports or case series with a small number of patients [35]. In a larger observational study, myasthenia had a prevalence of only 0.5% (2 patients out of 439) [10]. Other muscles and neuromuscular-junction abnormalities linked to the diagnosis of COVID-19 were myalgias, myositis, and dermatomyositis [31].

#### 3.1.4. Sensorial Disorders and other Cranial Nerve Dysfunctions

We included sensorial abnormalities secondary to SARS-CoV-2 infection under a separate heading due to their clinical significance in the acute viral disease.Anosmia (total loss of smell), hyposmia (partial loss of smell), and rarely parosmia (abnormal perception of an olfactory stimulus) and their taste-disorder counterparts—ageusia, hypogeusia, and parageusia—were recognized early in the COVID-19 pandemic as common symptoms highly suggestive of this etiology. The reported prevalence of these findings varied considerably between 7% and 80% over the course of the pandemic [3,5,10,31,36]. In the early stages, one of the first Chinese studies reported a prevalence of 5% [37]. Subsequently, another study collecting data from a German population reported that smell and taste impairment were much more common [38]. In a comprehensive systematic review including more than 38,000 patients, von Bartheld et al. reported that sensorial impairment occurred frequently, in 43% of cases for smell impairment and 44.6% of cases for taste impairment [38,39]. The authors described important epidemiologic differences in various ethnic groups, independent of the severity of the viral infection. The prevalence of smell and taste abnormalities in Caucasians was almost three times higher than in Asian patients [39]. Another difference was related to the SARS-CoV-2 genotype: the wild-type (Wuhan) variant was associated with the highest prevalence of sensorial disturbances, whereas the Delta and Omicron variants with a lower prevalence [40]. Smell and taste disorders were overlapping most frequently (in up to 43% of cases) but could also be encountered as separate symptoms in up to 22% of cases [10,38,39].Typically, anosmia develops suddenly at a median of 7 days after COVID-19 onset and persists for 7 to 10 days. Rarely, smell and taste abnormalities can persist for months, improving progressively. The olfactory impairment is more severe and frequent in the elderly and can result in long-term disability in this population [36]. Besides anosmia/hyposmia and ageusia/hypogeusia, qualitative sensorial disturbances are more difficult to recognize. Parosmia or parageusia and phantosmia and phantageusia are sensorial perceptions in the absence of stimuli. Ercoli et al. identified phantosmia and phantageusia in up to one quarter of patients who recovered after COVID-19 [41].Other reported cranial nerve dysfunctions are oculomotor abnormalities, ocular neuropathies, facial palsy, hearing loss, and lower cranial-nerve dysfunction. Hearing loss was reported relatively frequently during or after COVID-19, with an incidence of 13% to 15% [5,31]. Sudden unilateral or bilateral neurosensorial hearing loss and persistent tinnitus have been described by patients [31]. The incidence of Bell’s palsy almost doubled in 2020 compared to the pre-pandemic period in the same location in Northern Italy [42]. Another indirect piece of evidence of the link between COVID-19 and facial palsy was added by a study that found unexpectedly high seropositivity for SARS-CoV-2 among patients with Bell’s palsy [43]. Isolated cranial-nerve dysfunctions were reported in up to 7% of COVID-19 cases [10].

### 3.2. Neuroimaging 

The imaging findings in patients with neuroCOVID are variable and depend on the underlying pathology. Various modalities such as CT, MRI, or PET were used to describe the radiological findings in patients infected with SARS-CoV-2 presenting neurological abnormalities.

In patients with cerebral ischemia, small acute infarcts, abnormal microangiopathy, and small areas of hemorrhage have been described on CT [44]. A variety of changes, including fronto-temporal hypoperfusion, frontal ischemia, and midbrain lesions, was recognized in patients with cognitive impairment and SARS-CoV-2 infection [45]. In those presenting with ischemia, all cerebral vessels can be affected by thromboembolic events. In cases with encephalitis, most studies found no abnormalities on CT or MRI [45]. However, positive non-specific findings were described sparingly in case reports: some case reports described hyperintensity signals of grey matter on MRI distributed in the mesial temporal lobe, hippocampus, and orbitofrontal cortex [46]. Acute necrotizing encephalopathy has been described in COVID-19 as multifocal lesions distributed predominantly in the thalamus and brainstem [46]. In patients with anosmia secondary to SARS-CoV-2, MRI studies revealed T2/FLAIR hyperintensity of the olfactory bulbs with or without olfactory-cleft edema [47]. In a series of COVID-19-associated paraspinal myositis, MRI found edema and enhancement in paraspinal muscles [48]. 

### 3.3. Pathophysiology

The pathophysiology of the neurological injury following acute COVID-19 infection is not entirely elucidated. It is likely that the virus can damage the neuronal tissue directly or through secondary injury (Figure 1). 

The direct invasion of the CNS leading to meningitis or encephalitis, entities that have been described in patients with COVID-19, could be an important pathogenic mechanism [49]. A systematic review including 14 studies found a prevalence of 6% of CSF positivity for SARS-CoV-2 [50]. At the same time, not all patients with the virus present in the CSF had a positive COVID-19 nasal swab, suggesting that the virus may have a different infective pattern in the CNS [50]. The SARS-CoV-2 virus can interact with neurons via the ACE II receptors presenting in CNS, especially in circumventricular organs, the subfornical organ, paraventricular nucleus, nucleus of the tractus solitaries, and rostral ventrolateral medulla [51]. It can also reach neurons by blood through a malfunctioning blood–brain barrier (BBB), by neuronal retrograde transport, through the olfactory epithelium and transcribrial plate into the CSF, and lymphatics [52,53]. Although poorly documented, CSF dissemination can be responsible for frontal-lobe injuries or brainstem with respiratory pattern dysfunction. Other studies identified markers of BBB destruction and inflammatory proteins but failed to find viral particles in the CSF [54]. 

A secondary neuronal injury could be induced by hypoxia, microthrombosis, or cytokine storm [45,55,56]. For example, proposed pathophysiological mechanisms for persistent fatigue, a common sequela of COVID-19, are cerebral hypoxemia, inflammation, and mitochondrial dysfunction. Some authors suggested that fatigue could also be a reflection of metabolic-reserve exhaustion [45]. 

These mechanisms seem to be of lesser importance given that hypoxia secondary to other non-viral causes of acute respiratory distress syndrome (ARDS) is not associated with neurological impairment. Microthrombosis seems to be induced by endothelial impairment and coagulation disturbances present in SARS-CoV-2 infection [56]. Brain endothelial-cell lesions could be mediated by the SARS-CoV-2 M^PRO^ protease that cleaves NF-kB essential modulator (NEMO), an important modulator of apoptosis and antiviral type I interferons. In a study on mice brains, NEMO dysfunction induced microvascular pathology [57]. 

Other types of systemic cytokine storm encountered in burns, acute pancreatitis, or trauma are not always associated with neurological features. The cytokine release during COVID-19 is not greater than in the conditions mentioned above. Another secondary injury could be mediated by glial-cell activity. Although these cells are primarily responsible for protecting the brain from systemic inflammation, they have been also implicated in mediating neuronal injury secondary to SARS-CoV-2 [55,58]. A maladaptive glial response like the one previously described in neuropsychiatric pathologies was hypothesized to act as a mediator for abnormal neurological COVID-19 aftermath [58]. Another connection between long neuroCOVID and mental-health disorders could be the functional impairment of the glial cells and the tau protein, known to occur in PTSD [59,60,61].

## 4. Conclusions

COVID-19 infection is frequently associated with neurological impairments that can range from mild to severe. The pathophysiology of COVID-19-related neuronal-tissue injury is not well understood; however, several mechanisms such as neuron direct injury by SARS-CoV-2, microvascular pathology, glial-cell dysfunction, hypoxia, and cytokine release have been reported as involved. In addition, the psychological stress conjoined with the one accompanying SARS-CoV-2 infection is likely an important determinant of brain injury.

Future studies are required to describe in depth the pathophysiology and risk factors for the neurological impairment associated with COVID-19. 

## Figures and Tables

**Figure 1 brainsci-12-01531-f001:**
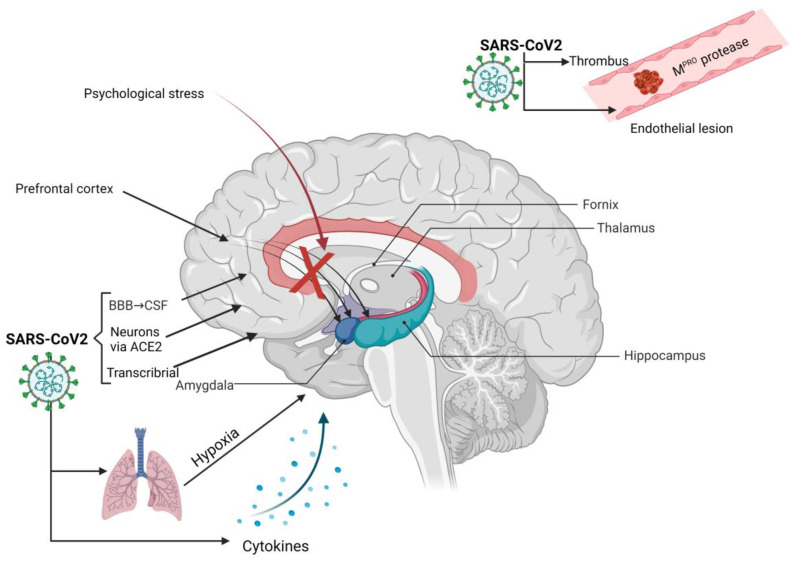
Main mechanisms involved in brain injury following COVID. *BBB*—blood–brain barrier, *CSF*—cerebrospinal fluid, *ACE2*—angiotensin-converting enzyme 2 receptor.

## Data Availability

Data used for this review are publicly available from the sources described in the methods section.

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
