# Peer review of "Neurological Manifestations of SARS-CoV2 Infection: A Narrative Review"

_brainsci, 2022, doi:10.3390/brainsci12111531_

Round 1
Reviewer 1 Report
The authors reported a narrative review about neurological manifestations of SARS-Cov2 infection. I have some comments to the authors:
- Please include in the title that the work is a “narrative review”.
- Please better describe the possible long-term sequelae of COVID-19
- In the paragraph related to neuropsychiatric symptoms, the authors missed and important and frequent condition in neurological practice such as Functional Motor Disorders. Here the reference for general audience about Functional Motor Disorders (1) and the case description of one case during covid-19 (2):
(1) Lidstone SC, et al. Functional movement disorder gender, age and phenotype study: a systematic review and individual patient meta-analysis of 4905 cases. J Neurol Neurosurg Psychiatry. 2022 Jun;93(6):609-616. doi: 10.1136/jnnp-2021-328462. Epub 2022 Feb 25. PMID: 35217516.
(2) Piscitelli D, et al. Functional movement disorders in a patient with COVID-19. Neurol Sci. 2020 Sep;41(9):2343-2344. doi: 10.1007/s10072-020-04593-1. Epub 2020 Jul 14. PMID: 32661885; PMCID: PMC7358315.
- Gustative impairment is also an important finding among patients with COVID-19
- The last two paragraph of “Pathophysiology” are not really necessary. I recommend the authors providing just the information, not even certain, about the pathophysiological explanation of neurological symptoms COVID-19 related.
Author Response
Thank you for your valuable suggestions regarding our manuscript.
Here are our answers to your suggestions:
-We included the narrative review wording in the title.
-We agree that long-term sequellae after neuroCOVID are an important part of the long-term COVID syndrome. While the literature provides little information in this area, we included more information on long-term outcomes in the manuscript.
-We added a new paragraph on the link between functional motor disorders and SARS-Cov2 infection. We also added more detailed information on the gustative impairment during COVID.
-We removed the last 2 paragraphs on pathophysiology, as suggested by the reviewer.

Reviewer 2 Report
The Review (brainsci-2013015), titled “Neurological manifestations of SARS-Cov2 infection” aimed to provide a descriptive analysis of correlations between the SARS-Cov2 infection and the central nervous system. In general, the Review is interesting and clear to read with high level of novelty.
Specific comments:
Line 27 in the Abstract Authors should better specify the time frame of analysis for example starting from 2019 to 2022.
Lines 33-36 In the description of the COVID-19 infection all neurological symptoms should be included and described on the text. In order to give a more comprehensive description Authors should consider that among COVID-19 neurological symptoms are indicated phantosmia, parosmia, phantogeusia, and parageusia as reported in a previous study (Ercoli et al., 2021, https://doi.org/10.1007/s10072-021-05611-6). In addition, it is important to note that these smell and/or taste qualitative disturbances were reported in 35% of patients who had COVID-19 acute infection.
Line 40 It should be " Expiritu and colleagues".
Line 72 The sentence “We did not include in this review grey literature” is not clear please try to explain this finding.
Line 87 As regards long-term symptoms Authors should indicate the presence of phantosmia, parosmia, phantogeusia, and parageusia as previously suggested.
Line 121 In the list of Nonspecific neurological/neuropsychiatric symptoms Authors should consider the presence of qualitative disturbances of olfactory and/or gustatory function in 35% of patients after COVID-19 acute infection.
My major concern is on the classification of the olfactory deficits since anosmia and hyposmia may show different etiologies. At the peripheral level anosmia and hyposmia could be associated to a damage of olfactory receptors, to a decrease of neurogenic processes, to exposure to toxic environmental agents, and to ossification and/or closure of foramina in the cribriform plate. Instead, at central level olfactory deficits could be associated to a decrease in olfactory bulb volume, to a decrease in the turnover of interneurons in the olfactory bulb and also to a reduced activity in the olfactory cortex. Moreover, parageusia and phantosmia are usually considered related to a central deficits.
Line 127 Authors should explain the ICU acronym.
Line 279 It should be "COVID-19".
Line 307 It should be "associated with".
Author Response
We want to thank the reviewer for his/her valuable comments on our manuscript.
-As suggested, we added more clarification on the olfactory and gustative disturbances during COVID-19 infection. We also included information on qualitative sensorial abnormalities.
-We clarified the meaning of 'grey literature' in the methods section.
-We included an explanation for the ICU acronym
-We made the other spelling modifications as suggested by the reviewer.

Round 2
Reviewer 2 Report
Authors revised the Manuscript according to the Reviewer's suggestion.
Author Response
Thank you. There are no futher chnages to the manuscript.